**Subject Category:**
Biology (whole organism)

behaviour/physiology

stress, coping style, proactive, reactive, cortisol, *Danio rerio*

**Author for correspondence:**
Ryan Y. Wong
e-mail: rwong@unomaha.edu

# Differences in stress reactivity between zebrafish with alternative stress coping styles

Ryan Y. Wong[1,2], Jeffrey French[1,2] and Jacalyn B. Russ[1]

[1]Department of Biology, and [2]Department of Psychology, University of Nebraska Omaha, Omaha, NE 68182, USA

RYW, 0000-0003-0236-672X

Animals experience stress in a variety of contexts and the behavioural and neuroendocrine responses to stress can vary among conspecifics. The responses across stressors often covary within an individual and are consistently different between individuals, which represent distinct stress coping styles (e.g. proactive and reactive). While studies have identified differences in peak glucocorticoid levels, less is known about how cortisol levels differ between stress coping styles at other time points of the glucocorticoid stress response. Here we quantified whole-body cortisol levels and stress-related behaviours (e.g. depth preference, movement) at time points representing the rise and recovery periods of the stress response in zebrafish lines selectively bred to display the proactive and reactive coping style. We found that cortisol levels and stress behaviours are significantly different between the lines, sexes and time points. Further, individuals from the reactive line showed significantly higher cortisol levels during the rising phase of the stress response compared with those from the proactive line. We also observed a significant correlation between individual variation of cortisol levels and depth preference but only in the reactive line. Our results show that differences in cortisol levels between the alternative stress coping styles extend to the rising phase of the endocrine stress response and that cortisol levels may explain variation in depth preferences in the reactive line. Differences in the timing and duration of cortisol levels may influence immediate behavioural displays and longer lasting neuromolecular mechanisms that modulate future responses.

## 1. Introduction

When encountering a stressor animals typically exhibit behavioural and neuroendocrine responses to adaptively respond to stress. Despite stressors occurring across disparate contexts (e.g. foraging,

predator inspection, exploration of novel environments), an animal's behavioural response to these stressors are often constrained and predictable [1–5]. These behavioural responses are part of a correlated suite of traits that show consistent between-individual differences across time and contexts (i.e. stress coping style, personality type) [1–5]. In parallel with behavioural responses, stressors also elicit a stereotypical neuroendocrine stress response that results in rising glucocorticoid levels before returning to baseline. While many studies have identified the behavioural displays that comprise a personality type [1–5], we are just starting to get a thorough understanding of the associated endocrine dynamics.

Proactive and reactive stress coping styles consist of two qualitatively different sets of correlated behavioural responses to overcome stress [1,4–7]. An individual with a proactive coping style displays more risk-prone behaviours across contexts such as being more aggressive, actively inspecting predators and exploring novel environments. Proactive individuals also rely on feed-forward memory processes that result in low behavioural flexibility [4,5]. By contrast, individuals with a reactive stress coping style are more risk-averse (e.g. less aggressive, predator inspection and exploratory tendencies). Reactive individuals are more sensitive to changes in the environment and can perform a wider range of behavioural responses (i.e. high behavioural flexibility).

The behavioural responses to stressors both impact and are impacted by the neuroendocrine stress axis. Individuals with either coping style show a typical hypothalamic–pituitary–adrenal (HPA) axis-mediated glucocorticoid response to a stressor where glucocorticoid levels in the blood rise, peak, and then return to baseline through a negative feedback loop [4,8]. Despite the classic physiological stress response seen in both coping styles, reactive individuals have a relatively higher peak glucocorticoid response compared with the proactive individuals [4,5,9,10]. However, peak cortisol level is only one stage in the temporal sequence of the stress response. Studies have demonstrated individual variation in cortisol levels at baseline, release rate and responsiveness of the negative feedback loop [11–17]. How the temporal dynamics of cortisol signalling differs between stress coping styles is only beginning to be understood [13,16]. Circulating cortisol concentrations also have been shown to vary with other factors including sex and recent behavioural experience, and such factors can contribute to individual variation in the temporal responses of glucocorticoids following the perception of a stressor [18–23].

Artificial selection is a powerful method for studying the proximate and ultimate mechanisms of personality types (i.e. stress coping styles) [7,24,25]. The frequency of occurrence and magnitude of the selected trait both diverges and increases in the population over several generations of selective breeding [10,26–28]. Other closely linked traits may also be indirectly selected. For example, great tits (*Parus major*) and rodents (*Mus musculus*, *Rattus norvegicus*) selectively bred for divergent latency to explore or attack, respectively (i.e. proactive and reactive) show consistently divergent behavioural and glucocorticoid responses in a variety of contexts [4,6,11,12,26,29–31]. Japanese quail (*Coturnix japonica*), zebra finch (*Taeniopygia guttata*) and rainbow trout (*Oncorhynchus mykiss*) selectively bred for divergent cortisol responses to a restraint stressor also show different behavioural responses consistent with proactive and reactive stress coping styles [5,9,10,31–38]. Hence, use of artificially selected lines allow for the investigation of behaviourally and physiologically linked traits.

In this study, we investigated the endocrine response in two zebrafish (*Danio rerio*) lines (high stationary behaviour (HSB) and low stationary behaviour (LSB)) that were artificially selected to display divergent amounts of stationary behaviour to a novelty stressor [28]. Despite the HSB and LSB lines showing behavioural, morphological and neuromolecular differences consistent with the reactive and proactive stress coping styles [28,39–43], it is not known whether these lines also show correlated differences in cortisol stress reactivity. Here we exposed zebrafish from the HSB and LSB lines to a standardized stressor, recorded behavioural responses during stress exposure and measured whole-body cortisol prior to and at two time points during stress exposure (one that represents the rising phase and another that represents the falling phase). We tested the prediction that in response to a novelty stressor the HSB line will have a greater stress reactivity relative to the LSB line. We also evaluated whether behavioural and cortisol responses to the stressor were correlated within and between the selected lines and whether these relationships were similar in males and females.

# 2. Methods

## 2.1. Subjects

We used zebrafish from lines selectively bred to display contrasting amounts of stationary behaviour (HSB, LSB) [28]. In brief, starting from approximately 200 wild-caught individuals from a village near

Gaighata, India, we generated these two zebrafish lines through artificial selection on the amount of stationary behaviour displayed in an open field test (see [28] for more details on selection paradigm). In each generation, we identified individuals displaying the most pronounced stationary behaviour in an open field test for each line and used them to generate the next generation of the line. We maintain both of these lines through the use of this bidirectional selective breeding paradigm. While we used response in the open field assay to generate and maintain our lines, we showed that on average individuals of the LSB line display significantly more risk-prone behaviours than those of the HSB line across five other distinct assays [28]. Of note the LSB line showed significantly less stationary time in the open field test and significantly more time in the top half of the water column in the novel tank diving test (NTDT) compared with the HSB line (see [28] for more details regarding consistent line differences observed in the other behavioural assays). Individuals of both lines also show high behavioural repeatability across time [39]. Selective breeding on a behavioural trait also resulted in individuals of the HSB line having a smaller caudal region and slower fast-start performance relative to the LSB line [40]. Neurotranscriptome profiling showed that both lines differ in neurometabolism and neuroplasticity, among other functions [41–43]. Collectively, individuals from the LSB line on average show more risk-prone behaviours and differ in morphological and neurotranscriptome characteristics linked to the proactive stress coping style (*sensu* [1,4,5]). Similarly, the HSB line display several traits consistent with the reactive stress coping style. Thus, for clarity we refer to any individual from the LSB line as being proactive (i.e. proactive line) and any from the HSB line as being reactive (i.e. reactive line) for the remainder of the text.

Individuals of both proactive and reactive lines in the current study underwent nine generations of selective breeding and were 18–24 months post-fertilization at time of testing. All fish were housed in mixed-sex 10-gallon tanks on a custom-built recirculating system. The fish were kept on a 14 : 10 L/D cycle with water temperature set at 26°C. Fish were fed twice daily with Tetramin Tropical Flakes (Tetra, USA).

## 2.2. Behavioural stress assay

We used the behaviourally, pharmacologically and neuroendocrinologically validated NTDT to induce a behavioural and physiological stress response to a novelty stressor [44–47]. Using established procedures in our laboratory [28,43], the NTDT assay involved placing individual zebrafish in a trapezoidal tank (28 × 9.5 × 15 cm tall) filled with 2 l of system water for either 6 or 30 min where the stressor is the novel environment. We video-recorded each animal for later behavioural analyses using commercial software (Ethovision XT, Noldus). Specifically, we virtually divided the tank into two zones representing the upper and lower halves of the water column. We then quantified the amount of time spent in the upper half of the water column and movement in the entire arena (total distance swam). Time spent at a particular depth and movement are common behavioural stress response measures and are both negatively correlated with stress levels in fish [43,45,48,49]. Our previous study using the same NTDT assay showed that individuals from the reactive line spent significantly less time in the upper zone and had a lower amount of locomotor behaviour compared with the proactive line [28].

To facilitate identifying a robust physiological stress response in each line, we screened for and only used individuals displaying pronounced depth preferences. More specifically, if a reactive line fish spent greater than 151 s of the first 6 min in the bottom zone, then it was immediately sacrificed afterwards for the 6-min time point ($n = 12$; 6 males and 6 females), or left in the NTDT for an additional 24 min ($n = 12$; 4 males and 8 females) for the 30-min time point and then immediately sacrificed. Fish from the proactive line followed the same procedure ($n = 12$ for each time point; 7 males and 5 females for the 30-min time point; 6 males and 6 females for the 6-min time point) except the animal must have spent greater than 151 s of the first 6 min in the upper zone. We determined the time criteria through a pilot study that involved behaviourally screening fish from both the proactive and reactive lines in a 6 min NTDT assay (electronic supplementary material, figure S1). In total, 20 and 195 behaviourally tested individuals from the reactive and proactive lines, respectively, did not meet the criteria. As studies have shown that cortisol levels and activity are positively correlated [21–23], we exposed a random set of fish from both lines ($n = 12$ from each line; 6 males and 6 females in each line) to the NTDT for 6 min and then sacrificed them. Animals caught directly from the home tank and then immediately sacrificed served as the baseline ($n = 12$ from each line; 5 males and 7 females for the proactive line; 4 males and 8 females for the reactive line). With previous studies showing that whole-body cortisol levels peak between 10–15 min post-stressor in zebrafish and then begin to return to baseline over the next 60 min [13,50–52], we chose the 6- and 30-min time points to represent the rising and recovery to

baseline phases of the stress response, respectively. The sacrificing procedure involved decapitation followed by flash freezing on dry ice. Prior to flash freezing of the body, we assigned sex of each individual through visualization of testes or ovaries on dissection. Bodies were then stored at $-80°C$ until cortisol extraction and quantification. Testing occurred in a 4-hour window during the middle period of the light phase of the photoperiod (6.5–10.5 h after light-onset). All procedures were approved by the Institutional Animal Care and Use Committee of University of Nebraska Omaha/ University of Nebraska Medical Center (17-070-00-FC, 17-064-08-FC).

## 2.3. Cortisol extraction and assay

Whole bodies were thawed, weighed and then homogenized in phosphate buffered saline (PBS) with a Bullet Blender Storm (Next Advance) using zirconium oxide beads. Cortisol was subsequently extracted from the tissue homogenates using established procedures [44,53]. In brief, we added 5.0 ml diethyl ether to each sample, centrifuged and then snap froze the aqueous phase in a dry ice-methanol bath. The ether supernatant was poured off and evaporated under compressed air. Samples were resuspended in 1 ml PBS and stored at $-20°C$ until assay.

We measured cortisol using a characterized enzyme immunoassay [54]. Standards ranged from 7.8 to 1000 pg per well. We assessed immunological validity of the assay for whole-body cortisol in zebrafish by assaying serial dilutions of multiple tissue homogenates. All samples produced displacement curves that did not differ from the slope of the curve produced by cortisol standards (electronic supplementary material, figure S2). Extracted samples were diluted 1 : 4 in PBS prior to assay, and standards, samples and a quality control sample were measured in duplicate on each plate. The interassay coefficient of variation ($n = 2$) was 3.7%, and the intra-assay coefficient of variation was 4.5%. Concentrations from the assay results were corrected by body mass to yield ng cortisol/g body mass.

## 2.4. Statistics

We performed all statistical analyses using SPSS (version 24). To assess changes in cortisol levels across time, we used a generalized linear model (GLZM). Line, sex and time were included as between-subjects variables. Within each line, there was no significant difference in cortisol levels between individuals with or without the behavioural screening criteria at the 6-min time point (GLZM Wald $\chi^2$: 0.762, $p = 0.383$). Therefore, in each line we combined all individuals into one group for the 6-min time point in the GLZM. We similarly assessed differences in behavioural measures (time in upper zone, and movement) in a GLZM with time, sex and line as between-subjects variables. To be able to compare between the two time points, we calculated the per cent of total trial spent in the upper zone and total distance travelled per minute of trial time for each fish. Significant main and interaction effects were probed further by assessing for simple main effects within the statistical model. We used Pearson correlations to assess relationships between behaviour (per cent of trial time in upper zone, total distance travelled per min) and cortisol levels for both lines combined and separately. To account for multiple comparisons, in all analyses, we applied a Benjamini–Hochberg correction to determine significance [55].

# 3. Results and discussion

## 3.1. Reactive stress coping line has a higher cortisol level during rising phase than proactive line

While the reactive line had significantly higher cortisol levels relative to the proactive line (Wald $\chi^2$: 4.548, $p = 0.033$), this main effect is completely driven by the response to the stressor (figure 1 and table 1). There were no significant differences in cortisol levels between lines at baseline ($p = 0.806$; figure 1 and table 1). These findings are consistent with studies examining other species of teleost, birds and rodents where individuals of the reactive stress coping style had significantly higher post-stressor cortisol levels [4,11,17,32,56–59]. With some studies showing basal differences in cortisol between animals displaying the alternative stress coping styles while others do not (including the present study) [4,5,11–13,15,17,56,58], it is likely that different basal cortisol levels between the proactive and reactive stress coping styles vary by species.

There was a significant line × time point interaction effect (Wald $\chi^2$: 8.339, $p = 0.015$) on cortisol levels. Each line displayed significantly higher cortisol levels at both 6 and 30 min time points relative

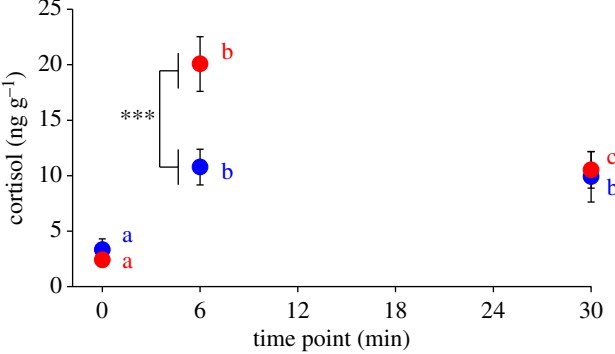

**Figure 1.** Interaction effect of line and time on whole-body cortisol levels. Only at the 6-min time point was there a significant difference in cortisol levels between lines (***$p < 0.001$). Different letters represent significant differences ($p \leq 0.05$) in whole-body cortisol levels between baseline, 6-min and 30-min time points within a line. Data are mean $\pm$ 1 s.e.m. Symbols and letters are colour-coded such that blue and red colours indicate the proactive and reactive lines, respectively.

**Table 1.** Statistical analyses of whole-body cortisol levels between time points.

| | GLZM contrast $p$-value for time point comparison | | |
|---|---|---|---|
| grouping | 0 versus 6 min | 0 versus 30 min | 6 versus 30 min |
| reactive and proactive | $1.36 \times 10^{-10}$ | 0.001 | 0.008 |
| reactive | $1.49 \times 10^{-10}$ | 0.006 | 0.001 |
| proactive | 0.006 | 0.053 | 0.623 |
| male | $7.71 \times 10^{-7}$ | 0.011 | 0.039 |
| female | $2.6 \times 10^{-5}$ | 0.066 | 0.053 |

to baseline (figure 1 and table 1). While there was a significant reduction in cortisol levels in the reactive line between the 6 and 30 min time points, this was not observed in the proactive line (figure 1 and table 1). Comparing between the lines, there was no significant difference in cortisol levels between the lines at baseline ($p = 0.806$) but the reactive line had significantly higher cortisol levels at 6 min than the proactive line ($p = 1.8 \times 10^{-5}$; figure 1). While we cannot rule out the possibility that the baseline group may represent behaviourally distinct individuals from the other two time points, we do not believe this impacts the endocrine response interpretations. Restricting our analysis to just the randomly selected individuals of each line at the baseline and 6-min time points, we similarly see (i) significantly higher cortisol levels in both the reactive ($t_{22} = -3.122$, $p = 0.051 \times 10^{-3}$) and proactive ($t_{22} = -3.122$, $p = 0.005$) lines compared with baseline and (ii) that the reactive line had significantly higher cortisol levels than the proactive line at the 6-min time point ($t_{22} = 2.487$, $p = 0.021$). As the 6-min time point represents the rising phase of the endocrine stress response, our data suggest that cortisol levels also differ between stress coping styles at time points outside of peak levels. One interpretation is that the reactive line may have a faster release rate (ng g$^{-1}$ cortisol per minute of stressor time) relative to the proactive line in the first 6 min post-stressor, which may lead to a shorter time to reach peak levels. However, it should be noted that our study design does not allow us to directly assess differences between lines in time to reach peak cortisol levels. Faster post-stressor corticosterone release has also been observed in selectively bred reactive great tits (slow-explorers) [11,12]. This suggests that glucocorticoid release rates in response to a stressor may be conserved in proactive and reactive stress coping styles among diverse species.

Looking at the time point during the recovery phase of the cortisol stress response (30 min), there was no significant difference in cortisol levels between the lines ($p = 0.537$, figure 1). One interpretation could be that mechanisms facilitating the return of cortisol levels to baseline are equally effective in both lines. While the significant line×stressor time interaction effect suggests an intriguing possibility that the reactive line has a more effective negative feedback mechanism on the HPA axis or glucocorticoid clearance rate, there are important considerations to this interpretation. Without knowing the time and magnitude of peak cortisol levels in each line, it is difficult to determine if the reactive line had a

faster or more efficient recovery towards baseline. It is possible that the reactive line reached peak levels sooner and with no significant difference at 30 min between the lines, it would suggest the reactive line had a less effective recovery phase compared with the proactive. Alternatively, if peak levels were higher in reactive compared with proactive individuals as has been documented in other studies [4,5,9,10], then the greater magnitude of reduction seen in the reactive line at the 30-min time point would suggest they have more efficient or faster recovery to baseline. Future studies that include more frequent time sampling during the endocrine stress response between these lines are needed to assess the more nuanced temporal dynamics. A related study examining endocrine response in zebrafish with alternative stress coping styles observed no difference in time to reach peak cortisol levels or the magnitude of peak cortisol levels [13]. However, proactive zebrafish had a quicker recovery to basal levels relative to reactive [13]. Differences in behavioural screening criteria and use of non-selected lines may explain the discrepancy between the current and previous study.

Through comparisons of results from the current and prior studies by others, we propose that across taxa the proactive and reactive stress coping styles differ in the temporal dynamics of the endocrine stress response. More specifically, we posit that reactive individuals have a significantly faster cortisol release rate than proactive individuals but have similar peak levels [11–14]. There will also be different glucocorticoid recovery rates to basal levels between the alternative stress coping styles. The direction of causality between stress coping style and glucocorticoid response is unclear. Some studies predict that the neuroendocrine system organizes the behavioural traits into correlated suites [12,15], whereas others hypothesize that the glucocorticoid responses are a consequence [4,6]. Our study provides indirect support for the latter hypothesis. Through bidirectional selection of stationary behaviour in response to a novelty stressor [28], we generated the reactive and proactive zebrafish lines that resulted in divergent cortisol levels at a time point during the rising phase of the endocrine response. However, we cannot rule out that we were simultaneously, but indirectly, selecting for cortisol levels. Further, it is possible that different stressor types or selective breeding on a physiological as opposed to a behavioural response to a stressor can lead to different endocrine temporal profiles between the alternative stress coping styles [9,10,27,34].

There are many potential fast-acting consequences of differing release and recovery rates of cortisol between the alternative stress coping styles such as modulating gene expression and neural plasticity, which ultimately impact cognition and behaviour. We have previously shown that our unstressed reactive and proactive zebrafish lines have different whole-brain transcriptome profiles that include genes linked to stress and anxiety-related behaviours (GABA, nonapeptide and glucocorticoid neurotransmission) [41,42]. While decreased basal levels of glucocorticoid receptor in the brain are known to reduce the efficiency of negative feedback on the HPA axis in response to a stressor [60,61], there are no baseline differences in glucocorticoid receptor mRNA expression between the lines [42]. However, a gene critical in the rise of glucocorticoids in response to stress (*crhr1* [62]) and two genes that inactivate cortisol (*hsd11b2* and *hsd20b2* [63,64]), show significantly higher basal expression in the brains of individuals form the reactive relative to proactive line [42]. This may explain the reactive line's higher cortisol levels during the rising phase of the endocrine response and suggests that the line may be molecularly primed to have a faster return towards basal cortisol levels. We speculate that both differential basal and stress-induced expression of genes in the glucocorticoid signalling pathway lead to the observed difference in cortisol levels between the lines at the 6-min time point. Of note, basal expression levels of some genes can predict the magnitude of stress coping behaviour and the direction of these correlations are not all congruent across the two lines [42]. These line-specific correlations between basal gene expression and behaviour may be further modulated through temporal availability of cortisol during the stress response. It is unknown how differences in the duration of cortisol levels may impact neural activity and subsequent gene expression to modulate stress coping behaviour and other cognitive processes. It has been proposed that discrete coping styles may be maintained in a population due to different fitness optima in variable environments [65–68] and we speculate that variation in the temporal dynamics of stress-induced glucocorticoid levels may also influence an individual's fitness by modulating current and future responses through neuromolecular changes in the brain. We are just beginning to understand how neural activity, endocrine system and the behaviour intersect in each coping style [7,34,42,56,69].

## 3.2. Cortisol levels vary by sex and time

Overall males had higher levels of whole-body cortisol than females (Wald $\chi^2$: 6.866, $p = 0.009$) but the main effects of sex on cortisol levels can be explained by the post-stressor responses (figure 2*a* and

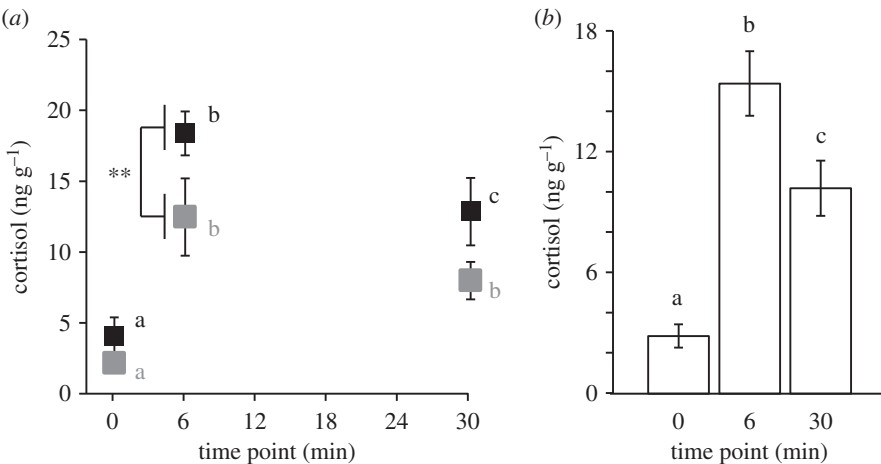

**Figure 2.** Effects of sex and time on whole-body cortisol levels. For (*a*) sex, there was a significant difference in cortisol levels between males and females only at the 6-min time point (**$p < 0.01$). Different letters represent significant differences ($p \leq 0.05$) in whole-body cortisol levels between baseline, 6-min and 30-min time points within sex. Symbols and letters are colour-coded such that black and grey colours indicate the males and females, respectively. For main effect of time (*b*) on cortisol levels different letters represent $p < 0.05$. Data are mean $\pm$ 1 s.e.m.

table 1). Males had significantly higher cortisol levels compared with females at the 6-min time point ($p = 0.006$) but there were no significant differences in cortisol levels between the sexes at baseline ($p = 0.572$) or the 30-min time point ($p = 0.098$; figure 2*a* and table 1). There was no significant interaction effect between sex and time (Wald $\chi^2$: 1.165, $p = 0.559$) or between sex, line and time on cortisol levels (Wald $\chi^2$: 0.272, $p = 0.873$). Our observation of higher stress-induced cortisol concentrations in males rather than females was not expected. Instead, female-biased elevation of post-stressor cortisol levels has been seen across several taxa [18–20]. We and others have previously shown that females tend to have higher stress- and anxiety-related behaviour relative to males in zebrafish, other teleosts, birds and mammals [10,28,70], which are associated with more robust endocrine responses in females [19,20]. To our knowledge, only two other studies have similarly examined sex differences in stress-induced cortisol levels of zebrafish and both show that males had significantly higher cortisol levels than females [71,72]. It is possible that sex-biased elevation in cortisol levels in response to a stressor is species- or context-specific [10,18–20,34,73] or may not be observed despite sex differences in corresponding behaviours [74].

The classic HPA-mediated neuroendocrine response to a stressor involves the elevation of cortisol levels that eventually trigger a negative feedback loop resulting in the return to basal levels. Not surprisingly, we observed this relationship on whole-body cortisol in our fish exposed to the novelty stressor (figure 2*b*, Wald $\chi^2$: 41.56, $p = 9.44 \times 10^{-10}$). Across both lines, cortisol levels at each time point (baseline, 6-min and 30-min) significantly differed from each other (figure 1 and table 1). Although we cannot rule out that our 6-min time point represented peak cortisol levels, prior studies have shown that cortisol levels peak between 10 and 15 min post-stressor in zebrafish [13,50]. Therefore, we believe the 6-min time point represents the rising phase of the glucocorticoid stress response. Similarly, cortisol levels at the 30-min time point are between both the 6-min time and basal levels (figure 1 and table 1), which suggests that cortisol levels are in the process of returning to basal levels.

## 3.3. Behavioural displays vary by coping style, sex and time

Behavioural responses to stress can also vary due to a variety of intrinsic and extrinsic mechanisms such as coping style, sex and context [1,3,4,68,75–78]. As expected, there was a significant main effect of line on the per cent total trial time spent in the upper zone (Wald $\chi^2$: 42.232, $p = 1.35 \times 10^{-10}$) and movement (Wald $\chi^2$: 26.018, $p = 3.38 \times 10^{-7}$) (figure 3). The significantly higher amount of time spent in the upper zone by the proactive line is a direct result of the behavioural screening criteria that we used. However, our prior study that randomly selected fish from each line showed that the proactive line spent significantly more time in the upper zone and had more movement than the reactive line [28]. The congruent observations of low stress-related behaviour and cortisol levels in response to the stressor

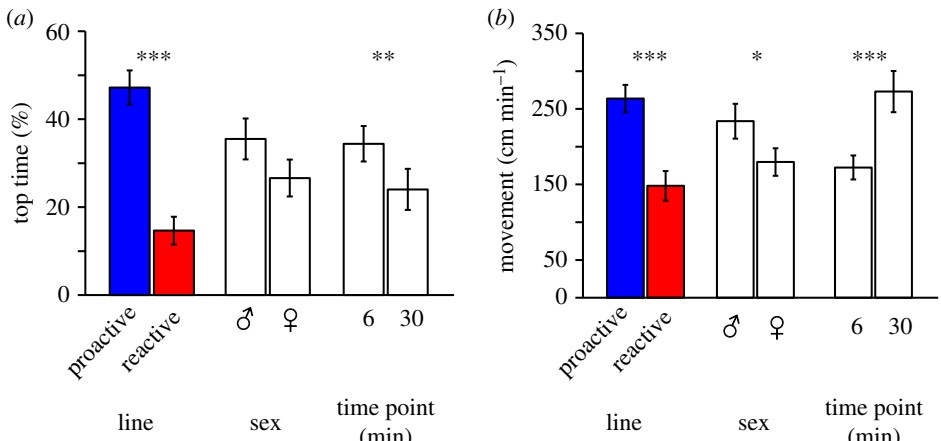

**Figure 3.** Effects of line, sex and time on behaviours. For per cent of trial time spent in the top zone (*a*), and movement (*b*): *$p < 0.05$; **$p < 0.01$; ***$p < 0.001$. Data are mean $\pm$ 1 s.e.m.

in the proactive line suggest that they have lower stress levels. Not surprisingly, we also found significant main effects of time on percentage of time spent in the upper zone (Wald $\chi^2$: 4.023, $p = 0.045$) and movement (Wald $\chi^2$: 19.213, $p = 1.2 \times 10^{-5}$) (figure 3). These changes in behaviour with stressor length may be due, in part, to habituation [79]. We did not observe any significant line × time interaction effects for time spent in the upper zone (Wald $\chi^2$: 0.292, $p = 0.589$) or movement (Wald $\chi^2$: 3.15, $p = 0.076$).

We did not observe a significant main effect of sex (Wald $\chi^2$: 1.379, $p = 0.24$) on time spent in the upper zone but did observe a sex effect on movement (Wald $\chi^2$: 4.799, $p = 0.028$; figure 3). Males showed significantly more movement compared with females ($p = 0.028$), which is consistent with another study examining these lines [28]. While other novelty stressor assays have documented sex differences in stress-related behaviours [28,70], to our knowledge, there have not been any reports of sex differences using the NTDT. We speculate that sex differences in stress and anxiety-related behaviours will depend on the focal behaviour and assay as seen in other contexts [10,18–20,28,34,73]. We did not observe any significant sex × time interaction effects for time spent in the upper zone (Wald $\chi^2$: 0.074, $p = 0.785$) or movement (Wald $\chi^2$: 1.235, $p = 0.266$).

## 3.4. Line-specific relationship between inter-individual variation of behaviour and cortisol levels

The current and prior studies have found group and line level differences between cortisol levels and zebrafish behaviour in the NTDT stressor assay [28,44,45,49]. Upon examination at the individual level, we did not observe any significant correlations between variation of whole-body cortisol and per cent of trial in the upper zone ($r = -0.025$, $p = 0.834$) or movement ($r = 0.06$, $p = 0.616$) for both lines combined (figure 4). When just analysing the proactive line there were no significant correlations between cortisol levels and per cent time in the upper zone ($r = 0.008$, $p = 0.964$) or movement ($r = 0.229$, $p = 0.18$; figure 4). However, in the reactive line there was a significant positive correlation between cortisol levels and per cent time in the upper zone ($r = 0.43$, $p = 0.009$; figure 4*a*) but not movement ($r = 0.26$, $p = 0.126$; figure 4*b*). These results indicate that whole-body cortisol levels and time spent in the top half of the water column show a more direct relationship with each other in only the reactive line. This suggests that individuals in the reactive line may be more responsive to small changes in cortisol levels and that cortisol amount is one mechanism explaining inter-individual variation in this behaviour in the NTDT. We hypothesize that the line-specificity of this relationship between cortisol and behaviour is part of a suite of mechanisms leading to overall increased cortisol reactivity in the reactive compared with the proactive line. It should be noted that a recent meta-analysis of correlational analyses between cortisol levels and behaviours in the NTDT suggest that such correlations are weak or non-significant [49]. Our results are generally consistent with this as we only observed a correlation between cortisol level and behaviour under specific parameters (e.g. one behaviour, one line). Relationships between cortisol levels and stress coping behaviours are complex and we cannot differentiate whether post-stressor cortisol levels can influence behaviours or if the stressor co-activates both behavioural and endocrine systems [15,34].

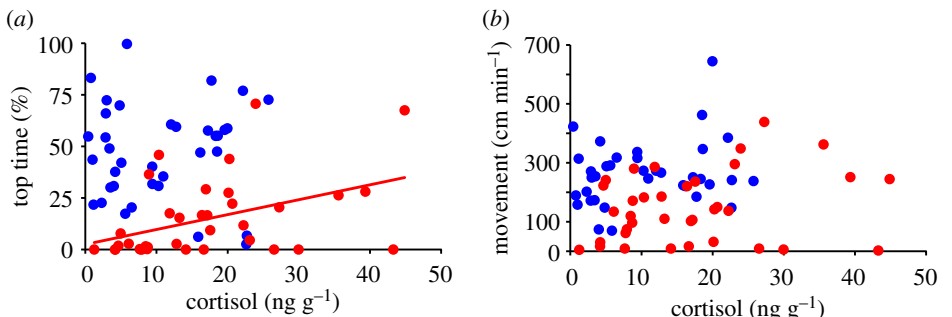

**Figure 4.** Correlation between whole-body cortisol levels and per cent time in upper zone (*a*) and movement (*b*). Trend line indicates significant correlation only for the reactive line in (*a*). Symbols are colour-coded such that blue and red colours indicate the proactive and reactive lines, respectively.

Despite studies showing that increasing cortisol levels are associated with increased locomotor activity in zebrafish and other teleost [21–23], locomotor activity does not likely explain our observations of line differences in cortisol levels. While we observed that the proactive line moved significantly more and had significantly lower cortisol levels relative to the reactive line, this pattern is opposite from observed differences in movement and cortisol levels of other studies [21–23]. There were also no significant correlations between movement and cortisol levels for either lines (figure 4*b*). Of note, just analysing fish that were randomly selected from each line and exposed to the NTDT for 6 min, the reactive line had significantly higher cortisol levels relative to proactive ($t_{22} = 2.487$, $p = 0.021$). This suggests that differences in cortisol levels in the NTDT represent trait differences between the lines and cannot be explained by state differences in movement in the tank or belong to a subpopulation of a line.

## 4. Conclusion

In response to a stressor, animals display behaviours and changes in physiology that are part of a suite of traits that are consistently different between individuals, and are stable within an individual across contexts and time. Two stress coping styles, proactive and reactive, comprise individuals that are risk-prone with low glucocorticoid response or risk-averse with higher glucocorticoid responses, respectively. However, not much is known about how cortisol levels differ between the alternative stress coping styles outside of peak levels. Our study highlights that bidirectional selection on a behavioural response to a novelty stressor leads to divergences in cortisol levels at a time point during the rising phase of the endocrine stress response. The reactive line, which was selected for low exploration (i.e. risk-averse), showed a significantly faster cortisol release rate within the first 6 min of being exposed to a novelty stressor compared with the proactive line. Further, inter-individual variation of a behavioural response to novelty stressor can be explained by variation in whole-body cortisol levels in only the reactive line. Prior studies characterizing the behavioural, morphological, neuromolecular and now the endocrine differences between the lines all support the observation that they represent the reactive and proactive stress coping styles. Differences in the timing and availability of cortisol during the stress response may lead to changes in frequency or duration of stress coping behaviours that impact the individual's survival. It is unclear how the elevation and recovery of cortisol levels in response to a stressor interact with neuromolecular mechanisms important in glucocorticoid signalling in the brain between the alternative stress coping styles. Future studies should examine how neural and transcriptional activity, behaviour and cortisol levels interact to gain a better understanding of how the proactive and reactive stress coping styles arise.

Ethics. All procedures were approved by the Institutional Animal Care and Use Committee of University of Nebraska Omaha/University of Nebraska Medical Center (17-070-00-FC, 17-064-08-FC).

Data accessibility. The dataset supporting this article is in the electronic supplementary material.

Authors' contributions. R.Y.W. conceived the study and performed statistical analyses. J.F. performed the enzyme immunoassays and its validation for zebrafish. J.B.R. conducted the experiment and performed data analyses. All authors contributed to the writing of the manuscript.

Competing interests. We have no competing interests.

Funding. Funding for this study was provided by National Institutes of Health (R15MH113074), Nebraska EPSCoR First Award (OIA-1557417), Nebraska Research Initiative and University of Nebraska Omaha start-up grants and Open Access Funds to R.Y.W. Funding was also provided by University of Nebraska Omaha Biology Department to J.B.R. Acknowledgements. We are grateful to D. Revers and A. Park for zebrafish husbandry. We thank J. Cavanaugh for help with the enzyme immunoassays. We thank M. Baker for helpful discussions and feedback on an earlier version of the manuscript.

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
