## [Reviewer comments · Royal Society Open Science]

Review History

RSOS-181797.R0 (Original submission)

Review form: Reviewer 1

Is the manuscript scientifically sound in its present form?

Yes

Are the interpretations and conclusions justified by the results?

Yes

Is the language acceptable?

Yes

Is it clear how to access all supporting data?

Yes

Do you have any ethical concerns with this paper?

No

Have you any concerns about statistical analyses in this paper?

No

Recommendation?

Accept with minor revision (please list in comments)

Comments to the Author(s)

Manuscript ID: RSOS-181797

Title: Differences in stress reactivity between zebrafish with alternative stress coping styles

Individuals vary in their behavioral responses to stressful conditions and, in some cases, may show consistent individual responses across contexts. The frameworks of 'behavioral syndromes' and 'animal personality,' predict that it may be possible to categorize animals qualitatively by consistent behavioral responses into 'proactive' and 'reactive' stress-coping styles. Despite these styles potentially representing only part of a broader spectrum of individual variation in behavior, and the expectation that such discrete behavioral phenotypes should only be expected when distinct adaptive behavioral peaks are present in the selective landscape, several studies have attempted to identify the genetic, neuroendocrine, or endocrine differences between fish expressing 'proactive' or 'reactive' behavioral phenotypes, with the ultimate goal of understanding the proximate mechanisms that underlie these phenotypic differences.

This manuscript describes the results of an experiment testing for differences in glucocorticoid stress reactivity profiles in 'proactive' and 'reactive' zebrafish (*Danio rerio*) using artificially-selected genetic lines that were bred to differ in behavioral responses to an acute stressor. Specifically, genetic lines had been selected to show either high or low stationary behaviors during an 'open field assay' test. In this study, the authors exposed these genetic lines of zebrafish to a 'Novel Tank' stressor test and then recorded their behavioral responses including the amount of time spent swimming/moving and the duration of time spent within the bottom or top half of the tank. The authors then used that behavioral data to identify a subset of fish from each genetic line with consistent responses based on those behavioral measures, and therefore could be categorized into discrete 'proactive' or 'reactive' coping styles; fish that spent more than 151 s of the first 6 min in the bottom zone of the tank were classified as 'reactive', while fish spending more than 151 s in the top zone of the tank were considered 'proactive.' The authors then sacrificed fish and measured whole-body cortisol levels to evaluate whether these 'reactive' and 'proactive' fish difference in glucocorticoid reactivity to the 'novel tank' test.

In brief, 'reactive' zebrafish were observed to have higher whole-body cortisol than 'proactive' fish 6 min after exposure to the 'novel tank' test. Since 'reactive' and 'proactive' fish no longer differed in whole-body cortisol 30 min after stressor exposure, these differences at 6 min suggest that the groups differ in HPI axis reactivity. The authors also observed a sex difference in whole-body cortisol response, with males zebrafish having higher whole-body cortisol concentrations at the 6 min sampling time following stressor exposure.

While the overall aim of this study - to test whether differences in the behavioral response to stressors are underlain by differences in reactivity of the HPI endocrine axis - the manuscript in its present form should be revised to clarify several issues. Those issues are outlined below.

Major Issues (listed in order of importance for revision)

1) Baseline group. Individual zebrafish sampled at the 6 min and 30 min time points to generate the stress reactivity data were subsets of the High Stationary Behavior (HSB) and Low Stationary Behavior (LSB) artificially-selected lines. Those subsets were fish identified as showing consistent

differences in depth preference in the 'novel tank' test (described in lines 118-126). Fish used for the baseline time point, however, were haphazardly selected from home tanks holding fish of the HSB and LSB lines (lines 131-133), and did not appear to be the same subset of fish classified as 'proactive' or 'reactive' based on the depth preferences. As such, the baseline fish are not a valid indicator of what baseline whole-body cortisol levels would be in the 'proactive' and 'reactive' groups, as they were likely to include a large scope of behavioral and (based on the author's hypothesis) neuroendocrine phenotypic variation.

Ideally, the authors would have used previously identified 'proactive' and 'reactive' fish for the baseline sample. Since this may not be possible at this time, the authors should revise Figures 1 and 2a to clearly indicate that the baseline fish are not equivalent to the 6 min and 30 min fish (perhaps by adding a dotted line between the baseline and 6 min samples, or gray shading the 0 min area of the plot). Similarly, the statistical analyses should be conducted for the 6 min and 30 min alone, and a separate analysis conducted for the two groups of baseline fish.

2) Clarity of HSB/LSB versus proactive/reactive phenotypes. The authors need to revise the manuscript to make the distinction between these phenotypes more clear to the reader. If I understand correctly, the genetic lines of zebrafish were artificially selected for HSB/LSB behaviors. The authors then are using those lines to identify individuals from each line showing 'extremes' of depth preference phenotypes following stress, and are referring to those phenotypes as "proactive" (for select fish coming from the LSB line) and "reactive" (for select fish coming from the HSB line). That needs to be made more clear to the reader in a revised version of the manuscript. I think that this could be done by revising the sentences describing the experimental design (Lines 82-93). Part of the confusion arises because the authors state that they are testing the HSB line versus the LSB line (see Lines 90-91, for instance), but are actually testing a particular phenotypic subset of the HSB line ("reactive" fish) versus a subset of LSB line ("proactive" fish).

3) Use of the term 'neuroendocrine.' At several places in the manuscript, the authors refer to their study examining "neuroendocrine responses" (see lines 24, 82, 199, 206, etc.). However, the current study looked at whole-body cortisol levels...which is not a neuroendocrine response, despite the HPI axis containing neuroendocrine signaling components such as CRH/CRF. This needs to be revised throughout the manuscript. The response variable measured in the present study should just be referred to as 'endocrine', given the data presented.

Minor comments:

- 1) Lines 20-21. For clarity, perhaps revise to the following: "Animals experience stress in a variety of contexts, and the behavioral and neuroendocrine responses to stress can vary among conspecifics."
- 2) Line 28. Which behaviors? Since the specific behavior of depth preference isn't described in the Abstract, the reader has a difficult time evaluating the relevance of this manuscript to their own interests/research.
- 3) Line 30. The term 'cortisol' release isn't really accurate here, as these are whole body cortisol measures and not circulating. Hence, cortisol released from the interrenal is indistinguishable from cortisol produced but not released. Please reword.
- 4) Line 42. Please add citations here.
- 5) Lines 68-69. The point made in this sentence should be made more explicitly: that circulating cortisol concentrations may be influenced by several factors. Perhaps revise to: "Circulating cortisol concentrations also have been shown to vary with other factors including sex and recent behavioral experience, and such factors can contribute to individual variation in the temporal responses of glucocorticoids following the perception of a stressor."
- 6) Lines 73-74. Revise to: "Other closely linked traits may also be selected indirectly."
- 7) Lines 80-81. This sentence is worded awkwardly. Please revise.

- 8) Line 83. Revise "selected to display divergent" to "selected for divergent"
- 9) Line 105. The stressor here in the Novel Tank Diving Test (NTDT) is just moving the fish to a novel tank, is that correct? Perhaps make this explicit in this sentence for readers unfamiliar with this test.
- 10) Lines 126-136. Please provide some number ($n = ?$) on how many fish were tested overall in order to identify the fish used for cortisol sampling (aka, how many fish from each the HSB and LSB lines were 'rejected' for not showing behaviors adhering to the "proactive" and "reactive" types as outlined in Lines 118-128?)
- 11) Lines 170-173. Were Pearson correlations run on all fish combined, or "reactive" and "proactive" groups separately? Or both?
- 12) Line 197. It's also possible that the reactive line shows a difference in glucocorticoid clearance rate. That should be mentioned as an alternative hypothesis here.
- 13) Line 206. The use of "neuroendocrine stress response" here should be revised to "HPA/HPI axis stress response", given that the present cortisol data and other studies being referenced refer to circulating glucocorticoids, and not hypothalamic-pituitary pathways.
- 14) Lines 206-208. This claim that reactive individuals have a fasted cortisol release rate doesn't seem to be well supported given the present data. Can the authors back this up further? Otherwise, the statement is too speculative based on the data presented to this point in the manuscript and should be revised. However, the date from reference [42], appears applicable here. Please revise.
- 15) Line 235. Delete the extra "be" between "further" and "modulated"
- 16) Lines 236-237. Revise to "...differences in the duration.."
- 17) Lines 238-239. This phrase about coping styles and different fitness optima is speculative, so should be worded as such. Perhaps revise to the following: "It has been proposed that discrete coping styles may be maintained in a population..."
- 18) Line 251. By "strain" do you mean "line"? I don't think this term has been used previously in the manuscript. For clarity, please use "line" here in Line 251 and in Line 301.
- 19) Lines 252-254. For clarity, revise to: "Our observation of higher stress-induced cortisol concentrations in males than females was not expected. Instead, female-based elevation of post-stressor cortisol levels has been seen across several taxa."
- 20) Lines 292-293. Do males of the HSB and LSB lines generally show greater movement? Is this phenotypic difference seen even in the absence of partitioning HSB fish into "reactive" and "non-reactive" HSBs? Similar for LSBs?
- 21) Figure 1. The authors need to indicate whether these are mean \pm SEM values, or \pm SD for this and the other figures.
- 22) Table 1. It needs to be stated somewhere that these are p values being presented.

Review form: Reviewer 2

Is the manuscript scientifically sound in its present form?

Yes

Are the interpretations and conclusions justified by the results?

Yes

Is the language acceptable?

Yes

Is it clear how to access all supporting data?

No

Do you have any ethical concerns with this paper?

No

Have you any concerns about statistical analyses in this paper?

No

Recommendation?

Accept with minor revision (please list in comments)

Comments to the Author(s)

I have completed my review of the manuscript titled "Differences in stress reactivity between zebrafish with alternative stress coping styles" by Wong et al. The authors have executed a brilliant, but simple, experiment that demonstrates differences in cortisol reactivity between two coping styles. If I had one criticism, I would have like to see physiological data related to more parts of the HPA axis (e.g., gene responses of cortisol receptors). This would help to identify what parts of the pathway have been changed (perhaps the authors are currently working on this). I typically do not have any "negative" major comments, but I am at a lost to find anything serious enough to warrant any. I do, however, have some minor comments that the authors should be able to address in a timely manner. My recommendation is to accept with minor revisions.

Minor comments

- 1) Review manuscript to ensure the past tense is being used. For example, line 82, "we investigate ..." where further in the paragraph the past tense is used.
- 2) Line 83 – it isn't clear to me the purpose. I think the confusing part of the sentence is the inclusion of the words after the colon.
- 3) More details needed on the HSB and LSB lines. For example, how long have they been divergent (i.e., how many generations). What were the original parents (wild type)? What was the original selection experiment that assigned fish to the two lines?
- 4) 80 references is a bit much for a data focused paper.
- 5) Line 184 – should a reference be included?
- 6) The authors mention exposing zebrafish to a novel stressor. Unless I'm missing it, I didn't see in the methods why was used as the stressor.

Review form: Reviewer 3

Is the manuscript scientifically sound in its present form?

Yes

Are the interpretations and conclusions justified by the results?

No

Is the language acceptable?

Yes

Is it clear how to access all supporting data?

Not Applicable

Do you have any ethical concerns with this paper?

No

Have you any concerns about statistical analyses in this paper?

No

Recommendation?

Reject

Comments to the Author(s)

In this manuscript, the authors quantified the whole-body cortisol levels in zebrafish that show different behaviours in novel tank diving test. The authors found differences in cortisol levels at 6 min between the animals that show „reactive“ vs. „proactive“ stress coping style as measured by the amount of time they spent in upper vs. lower zones in a novel tank. The authors also measured the differences in male vs. female zebrafish in their cortisol response.

The study addresses interesting question of the relationship between neuroendocrine stress response and coping style. While the question itself is interesting, I do have several major issues with this manuscript.

1. The authors conclude „differences in cortisol levels between the alternative stress coping styles are not just restricted to magnitude but also extends to the temporal dynamics.“ However the authors only analysed 3 time points, 0, 6, and 30 min and showed that there is a difference only at 6. In so far as the temporal dynamics is the only thing that is measured in this manuscript, it would be important for authors to show more data points to demonstrate the actual temporal dynamics. They need to complete the time points at 12, 18, and 24 in addition to 6 and 30 min to show the peak values and the rate of decrease.
2. One important question is whether different cortisol temporal dynamics shown in this manuscript is a general feature of selectively bred lines with different coping styles regardless of stressor type. Authors should test the temporal dynamic of the cortisol response of the lines to at least another stressor.
3. How reactive and proactive lines defined by the authors other behavioural criteria (such as open field test) fare in the NTDT need to be shown. Do reactive lines always show significantly more bottom-dwelling behaviour compared to the proactive lines? For this study, the authors chose a reactive line fish if it spent greater than 151 seconds of the first 6 minutes. They also need to show the cortisol temporal profile of reactive line fish that may not show such a pronounced behaviour. Such a fish may classify as “reactive” according to other behavioural criteria. In short a more comprehensive analysis is required to compare how reactive and proactive fish line behaves overall in NTDT test. Also the authors need to compare the cortisol response of different reactive and proactive fish and their behaviour in the NTDT not only the ones that show most pronounced behavioural differences in novel tank.
4. Somewhat related to the point 3, both in abstract and in the text the authors mention that they couldn't find correlations between individual variation of cortisol levels and behaviour, but this important point is not shown. The authors need to show this data and address the point of individual variation.

Decision letter (RSOS-181797.R0)

20-Feb-2019

Dear Dr Wong,

The editors assigned to your paper ("Differences in stress reactivity between zebrafish with

alternative stress coping styles") have now received comments from reviewers. We would like you to revise your paper in accordance with the referee and Associate Editor suggestions which can be found below (not including confidential reports to the Editor). Please note this decision does not guarantee eventual acceptance.

Please submit a copy of your revised paper before 15-Mar-2019. Please note that the revision deadline will expire at 00.00am on this date. If we do not hear from you within this time then it will be assumed that the paper has been withdrawn. In exceptional circumstances, extensions may be possible if agreed with the Editorial Office in advance. We do not allow multiple rounds of revision so we urge you to make every effort to fully address all of the comments at this stage. If deemed necessary by the Editors, your manuscript will be sent back to one or more of the original reviewers for assessment. If the original reviewers are not available, we may invite new reviewers.

- Data accessibility

<http://datadryad.org/submit?journalID=RSOS&manu=RSOS-181797>

- Competing interests

- Authors' contributions

- Acknowledgements

- Funding statement

on behalf of Dr Ryan Earley (Associate Editor) and Kevin Padian (Subject Editor)
openscience@royalsociety.org

Associate Editor's comments (Dr Ryan Earley):

Thanks for the submission, though please accept our apologies for the unusual delay in completing peer review: we struggled to find suitable referees, which imposed a delay. We're sorry for any inconvenience caused.

Nevertheless, we've now received three reviews of your manuscript. Two are broadly positive, while one is more critical. To give the authors the benefit of the doubt, while encouraging you to fully incorporate the changes requested (or to provide a solid scientific rebuttal if you choose not to do so), the Editors are recommending a revision of the paper. When you submit the revision, this will be returned to at least the more critical reviewer for a further assessment, so we urge you to make your changes clear and provide rationale for any that are not included. Good luck and we look forward to receiving the revision in the near future.

Comments to Author:

Reviewers' Comments to Author:

Reviewer: 1

Comments to the Author(s)

Manuscript ID: RSOS-181797

Title: Differences in stress reactivity between zebrafish with alternative stress coping styles

Individuals vary in their behavioral responses to stressful conditions and, in some cases, may show consistent individual responses across contexts. The frameworks of 'behavioral syndromes' and 'animal personality,' predict that it may be possible to categorize animals qualitatively by consistent behavioral responses into 'proactive' and 'reactive' stress-coping styles. Despite these styles potentially representing only part of a broader spectrum of individual variation in behavior, and the expectation that such discrete behavioral phenotypes should only be expected when distinct adaptive behavioral peaks are present in the selective landscape, several studies have attempted to identify the genetic, neuroendocrine, or endocrine differences between fish expressing 'proactive' or 'reactive' behavioral phenotypes, with the ultimate goal of understanding the proximate mechanisms that underlie these phenotypic differences.

This manuscript describes the results of an experiment testing for differences in glucocorticoid stress reactivity profiles in 'proactive' and 'reactive' zebrafish (*Danio rerio*) using artificially-selected genetic lines that were bred to differ in behavioral responses to an acute stressor. Specifically, genetic lines had been selected to show either high or low stationary behaviors during an 'open field assay' test. In this study, the authors exposed these genetic lines of zebrafish to a 'Novel Tank' stressor test and then recorded their behavioral responses including the amount of time spent swimming/moving and the duration of time spent within the bottom or top half of the tank. The authors then used that behavioral data to identify a subset of fish from each genetic line with consistent responses based on those behavioral measures, and therefore could be categorized into discrete 'proactive' or 'reactive' coping styles; fish that spent more than 151 s of the first 6 min in the bottom zone of the tank were classified as 'reactive', while fish spending more than 151 s in the top zone of the tank were considered 'proactive.' The authors then sacrificed fish and measured whole-body cortisol levels to evaluate whether these 'reactive' and 'proactive' fish difference in glucocorticoid reactivity to the 'novel tank' test.

In brief, 'reactive' zebrafish were observed to have higher whole-body cortisol than 'proactive' fish 6 min after exposure to the 'novel tank' test. Since 'reactive' and 'proactive' fish no longer differed in whole-body cortisol 30 min after stressor exposure, these differences at 6 min suggest that the groups differ in HPI axis reactivity. The authors also observed a sex difference in whole-body cortisol response, with males zebrafish having higher whole-body cortisol concentrations at the 6 min sampling time following stressor exposure.

While the overall aim of this study - to test whether differences in the behavioral response to stressors are underlain by differences in reactivity of the HPI endocrine axis - the manuscript in its present form should be revised to clarify several issues. Those issues are outlined below.

Major Issues (listed in order of importance for revision)

1) Baseline group. Individual zebrafish sampled at the 6 min and 30 min time points to generate the stress reactivity data were subsets of the High Stationary Behavior (HSB) and Low Stationary Behavior (LSB) artificially-selected lines. Those subsets were fish identified as showing consistent differences in depth preference in the 'novel tank' test (described in lines 118-126). Fish used for the baseline time point, however, were haphazardly selected from home tanks holding fish of the

HSB and LSB lines (lines 131-133), and did not appear to be the same subset of fish classified as 'proactive' or 'reactive' based on the depth preferences. As such, the baseline fish are not a valid indicator of what baseline whole-body cortisol levels would be in the 'proactive' and 'reactive' groups, as they were likely to include a large scope of behavioral and (based on the author's hypothesis) neuroendocrine phenotypic variation.

Ideally, the authors would have used previously identified 'proactive' and 'reactive' fish for the baseline sample. Since this may not be possible at this time, the authors should revise Figures 1 and 2a to clearly indicate that the baseline fish are not equivalent to the 6 min and 30 min fish (perhaps by adding a dotted line between the baseline and 6 min samples, or gray shading the 0 min area of the plot). Similarly, the statistical analyses should be conducted for the 6 min and 30 min alone, and a separate analysis conducted for the two groups of baseline fish.

2) Clarity of HSB/LSB versus proactive/reactive phenotypes. The authors need to revise the manuscript to make the distinction between these phenotypes more clear to the reader. If I understand correctly, the genetic lines of zebrafish were artificially selected for HSB/LSB behaviors. The authors then are using those lines to identify individuals from each line showing 'extremes' of depth preference phenotypes following stress, and are referring to those phenotypes as "proactive" (for select fish coming from the LSB line) and "reactive" (for select fish coming from the HSB line). That needs to be made more clear to the reader in a revised version of the manuscript. I think that this could be done by revising the sentences describing the experimental design (Lines 82-93). Part of the confusion arises because the authors state that they are testing the HSB line versus the LSB line (see Lines 90-91, for instance), but are actually testing a particular phenotypic subset of the HSB line ("reactive" fish) versus a subset of LSB line ("proactive" fish).

3) Use of the term 'neuroendocrine.' At several places in the manuscript, the authors refer to their study examining "neuroendocrine responses" (see lines 24, 82, 199, 206, etc.). However, the current study looked at whole-body cortisol levels...which is not a neuroendocrine response, despite the HPI axis containing neuroendocrine signaling components such as CRH/CRF. This needs to be revised throughout the manuscript. The response variable measured in the present study should just be referred to as 'endocrine', given the data presented.

Minor comments:

- 1) Lines 20-21. For clarity, perhaps revise to the following: "Animals experience stress in a variety of contexts, and the behavioral and neuroendocrine responses to stress can vary among conspecifics."
- 2) Line 28. Which behaviors? Since the specific behavior of depth preference isn't described in the Abstract, the reader has a difficult time evaluating the relevance of this manuscript to their own interests/research.
- 3) Line 30. The term 'cortisol' release isn't really accurate here, as these are whole body cortisol measures and not circulating. Hence, cortisol released from the interrenal is indistinguishable from cortisol produced but not released. Please reword.
- 4) Line 42. Please add citations here.
- 5) Lines 68-69. The point made in this sentence should be made more explicitly: that circulating cortisol concentrations may be influenced by several factors. Perhaps revise to: "Circulating cortisol concentrations also have been shown to vary with other factors including sex and recent behavioral experience, and such factors can contribute to individual variation in the temporal responses of glucocorticoids following the perception of a stressor."
- 6) Lines 73-74. Revise to: "Other closely linked traits may also be selected indirectly."
- 7) Lines 80-81. This sentence is worded awkwardly. Please revise.
- 8) Line 83. Revise "selected to display divergent" to "selected for divergent"

- 9) Line 105. The stressor here in the Novel Tank Diving Test (NTDT) is just moving the fish to a novel tank, is that correct? Perhaps make this explicit in this sentence for readers unfamiliar with this test.
- 10) Lines 126-136. Please provide some number ($n = ?$) on how many fish were tested overall in order to identify the fish used for cortisol sampling (aka, how many fish from each the HSB and LSB lines were 'rejected' for not showing behaviors adhering to the "proactive" and "reactive" types as outlined in Lines 118-128?)
- 11) Lines 170-173. Were Pearson correlations run on all fish combined, or "reactive" and "proactive" groups separately? Or both?
- 12) Line 197. It's also possible that the reactive line shows a difference in glucocorticoid clearance rate. That should be mentioned as an alternative hypothesis here.
- 13) Line 206. The use of "neuroendocrine stress response" here should be revised to "HPA/HPI axis stress response", given that the present cortisol data and other studies being referenced refer to circulating glucocorticoids, and not hypothalamic-pituitary pathways.
- 14) Lines 206-208. This claim that reactive individuals have a fasted cortisol release rate doesn't seem to be well supported given the present data. Can the authors back this up further? Otherwise, the statement is too speculative based on the data presented to this point in the manuscript and should be revised. However, the date from reference [42], appears applicable here. Please revise.
- 15) Line 235. Delete the extra "be" between "further" and "modulated"
- 16) Lines 236-237. Revise to "...differences in the duration.."
- 17) Lines 238-239. This phrase about coping styles and different fitness optima is speculative, so should be worded as such. Perhaps revise to the following: "It has been proposed that discrete coping styles may be maintained in a population..."
- 18) Line 251. By "strain" do you mean "line"? I don't think this term has been used previously in the manuscript. For clarity, please use "line" here in Line 251 and in Line 301.
- 19) Lines 252-254. For clarity, revise to: "Our observation of higher stress-induced cortisol concentrations in males than females was not expected. Instead, female-based elevation of post-stressor cortisol levels has been seen across several taxa."
- 20) Lines 292-293. Do males of the HSB and LSB lines generally show greater movement? Is this phenotypic difference seen even in the absence of partitioning HSB fish into "reactive" and "non-reactive" HSBs? Similar for LSBs?
- 21) Figure 1. The authors need to indicate whether these are mean \pm SEM values, or \pm SD for this and the other figures.
- 22) Table 1. It needs to be stated somewhere that these are p values being presented.

Reviewer: 2

Comments to the Author(s)

I have completed my review of the manuscript titled "Differences in stress reactivity between zebrafish with alternative stress coping styles" by Wong et al. The authors have executed a brilliant, but simple, experiment that demonstrates differences in cortisol reactivity between two coping styles. If I had one criticism, I would have like to see physiological data related to more parts of the HPA axis (e.g., gene responses of cortisol receptors). This would help to identify what parts of the pathway have been changed (perhaps the authors are currently working on this). I typically do not have any "negative" major comments, but I am at a lost to find anything serious enough to warrant any. I do, however, have some minor comments that the authors should be able to address in a timely manner. My recommendation is to accept with minor revisions.

Minor comments

- 1) Review manuscript to ensure the past tense is being used. For example, line 82, "we investigate ..." where further in the paragraph the past tense is used.

- 2) Line 83 – it isn't clear to me the purpose. I think the confusing part of the sentence is the inclusion of the words after the colon.
- 3) More details needed on the HSB and LSB lines. For example, how long have they been divergent (i.e., how many generations). What were the original parents (wild type)? What was the original selection experiment that assigned fish to the two lines?
- 4) 80 references is a bit much for a data focused paper.
- 5) Line 184 – should a reference be included?
- 6) The authors mention exposing zebrafish to a novel stressor. Unless I'm missing it, I didn't see in the methods why was used as the stressor.

Reviewer: 3

Comments to the Author(s)

In this manuscript, the authors quantified the whole-body cortisol levels in zebrafish that show different behaviours in novel tank diving test. The authors found differences in cortisol levels at 6 min between the animals that show „reactive“ vs. „proactive“ stress coping style as measured by the amount of time they spent in upper vs. lower zones in a novel tank. The authors also measured the differences in male vs. female zebrafish in their cortisol response.

The study addresses interesting question of the relationship between neuroendocrine stress response and coping style. While the question itself is interesting, I do have several major issues with this manuscript.

1. The authors conclude „differences in cortisol levels between the alternative stress coping styles are not just restricted to magnitude but also extends to the temporal dynamics.“ However the authors only analysed 3 time points, 0, 6, and 30 min and showed that there is a difference only at 6. In so far as the temporal dynamics is the only thing that is measured in this manuscript, it would be important for authors to show more data points to demonstrate the actual temporal dynamics. They need to complete the time points at 12, 18, and 24 in addition to 6 and 30 min to show the peak values and the rate of decrease.
2. One important question is whether different cortisol temporal dynamics shown in this manuscript is a general feature of selectively bred lines with different coping styles regardless of stressor type. Authors should test the temporal dynamic of the cortisol response of the lines to at least another stressor.
3. How reactive and proactive lines defined by the authors other behavioural criteria (such as open field test) fare in the NTDT need to be shown. Do reactive lines always show significantly more bottom-dwelling behaviour compared to the proactive lines? For this study, the authors chose a reactive line fish if it spent greater than 151 seconds of the first 6 minutes. They also need to show the cortisol temporal profile of reactive line fish that may not show such a pronounced behaviour. Such a fish may classify as “reactive” according to other behavioural criteria. In short a more comprehensive analysis is required to compare how reactive and proactive fish line behaves overall in NTDT test. Also the authors need to compare the cortisol response of different reactive and proactive fish and their behaviour in the NTDT not only the ones that show most pronounced behavioural differences in novel tank.
4. Somewhat related to the point 3, both in abstract and in the text the authors mention that they couldn't find correlations between individual variation of cortisol levels and behaviour, but this important point is not shown. The authors need to show this data and address the point of individual variation.

Author's Response to Decision Letter for (RSOS-181797.R0)

See Appendix A.

RSOS-181797.R1 (Revision)

Review form: Reviewer 1

Is the manuscript scientifically sound in its present form?

Yes

Are the interpretations and conclusions justified by the results?

Yes

Is the language acceptable?

Yes

Is it clear how to access all supporting data?

Yes

Do you have any ethical concerns with this paper?

No

Have you any concerns about statistical analyses in this paper?

No

Recommendation?

Accept as is

Comments to the Author(s)

Overall, the authors made appropriate revisions to the manuscript, and the major issues that arose with the prior version of the manuscript have been addressed. I have only a few small corrections for the authors with the revised version of their manuscript:

Lines 82-83. Please list for common names for each taxon here (Japanese quail, rainbow trout)

Line 236. 'great tits' shouldn't be hyphenated.

Line 371. Revise to "suggests"

Line 386. Revise to "reactive line"

Review form: Reviewer 2

Is the manuscript scientifically sound in its present form?

Yes

Are the interpretations and conclusions justified by the results?

Yes

Is the language acceptable?

Yes

Is it clear how to access all supporting data?

Yes

Do you have any ethical concerns with this paper?

No

Have you any concerns about statistical analyses in this paper?

No

Recommendation?

Accept as is

Comments to the Author(s)

None

Decision letter (RSOS-181797.R1)

24-Apr-2019

Dear Dr Wong:

On behalf of the Editors, I am pleased to inform you that your Manuscript RSOS-181797.R1 entitled "Differences in stress reactivity between zebrafish with alternative stress coping styles" has been accepted for publication in Royal Society Open Science subject to minor revision in accordance with the referee suggestions. Please find the referees' comments at the end of this email.

The reviewers and Subject Editor have recommended publication, but also suggest some minor revisions to your manuscript. Therefore, I invite you to respond to the comments and revise your manuscript.

- Ethics statement

- Data accessibility

<http://datadryad.org/submit?journalID=RSOS&manu=RSOS-181797.R1>

- Competing interests

- Authors' contributions

- Acknowledgements

- Funding statement

Because the schedule for publication is very tight, it is a condition of publication that you submit the revised version of your manuscript before 03-May-2019. Please note that the revision deadline will expire at 00.00am on this date. If you do not think you will be able to meet this date please let me know immediately.

on behalf of Dr Ryan Earley (Associate Editor) and Kevin Padian (Subject Editor)
openscience@royalsociety.org

Reviewer comments to Author:

Reviewer: 2

Comments to the Author(s)

None

Reviewer: 1

Comments to the Author(s)

Overall, the authors made appropriate revisions to the manuscript, and the major issues that arose with the prior version of the manuscript have been addressed. I have only a few small corrections for the authors with the revised version of their manuscript:

Lines 82-83. Please list for common names for each taxon here (Japanese quail, rainbow trout)

Line 236. 'great tits' shouldn't be hyphenated.

Line 371. Revise to "suggests"

Line 386. Revise to "reactive line"

Author's Response to Decision Letter for (RSOS-181797.R1)

See Appendix B.

Decision letter (RSOS-181797.R2)

25-Apr-2019

Dear Dr Wong,

I am pleased to inform you that your manuscript entitled "Differences in stress reactivity between zebrafish with alternative stress coping styles" is now accepted for publication in Royal Society Open Science.

on behalf of Dr Ryan Earley (Associate Editor) and Kevin Padian (Subject Editor)
openscience@royalsociety.org

Follow Royal Society Publishing on Twitter: [@RSocPublishing](https://twitter.com/RSocPublishing)
Follow Royal Society Publishing on Facebook:
<https://www.facebook.com/RoyalSocietyPublishing.FanPage/>
Read Royal Society Publishing's blog: <https://blogs.royalsociety.org/publishing/>

Appendix A

Response to Reviewer Comments

ID: RSOS-181797

Dear Editor

We are grateful for all of the constructive feedback from yourself and the reviewers. We acknowledge the common remarks regarding lack of clarity in several areas and that we should limit conclusions and interpretations to what was actually tested. We have made substantial revisions regarding these and other points the reviewers brought up. We believe our manuscript is even stronger and hope you find our revised version acceptable. We again thank the reviewers for their time and have addressed their feedback (in bold below) as well as in the revised manuscript.

Sincerely,

Ryan Wong, Jeffrey French, and Jacalyn Russ

Reviewers' Comments to Author:

Reviewer: 1

Comments to the Author(s)

Manuscript ID: RSOS-181797

Title: Differences in stress reactivity between zebrafish with alternative stress coping styles

Individuals vary in their behavioral responses to stressful conditions and, in some cases, may show consistent individual responses across contexts. The frameworks of 'behavioral syndromes' and 'animal personality,' predict that it may be possible to categorize animals qualitatively by consistent behavioral responses into 'proactive' and 'reactive' stress-coping styles. Despite these styles potentially representing only part of a broader spectrum of individual variation in behavior, and the expectation that such discrete behavioral phenotypes should only be expected when distinct adaptive behavioral peaks are present in the selective landscape, several studies have attempted to identify the genetic, neuroendocrine, or endocrine differences between fish expressing 'proactive' or 'reactive' behavioral phenotypes, with the ultimate goal of understanding the proximate mechanisms that underlie these phenotypic differences.

This manuscript describes the results of an experiment testing for differences in glucocorticoid stress reactivity profiles in 'proactive' and 'reactive' zebrafish (*Danio rerio*) using artificially-selected genetic lines that were bred to differ in behavioral responses to an acute stressor. Specifically, genetic lines had been selected to show either high or low stationary behaviors during an 'open field assay' test. In this study, the authors exposed these genetic lines of zebrafish to a 'Novel Tank' stressor test and then recorded their behavioral responses including the amount of time spent swimming/moving and the duration of time spent within the bottom or top half of the tank. The authors then used that behavioral data to identify a subset of fish from each genetic line with consistent responses based on those behavioral measures, and therefore could be categorized into discrete 'proactive' or 'reactive' coping styles; fish that spent more than 151 s of the first 6 min in the bottom zone of the tank were classified as 'reactive', while fish spending more than 151 s in the top zone of the tank were considered 'proactive.'

The authors then sacrificed fish and measured whole-body cortisol levels to evaluate whether these 'reactive' and 'proactive' fish difference in glucocorticoid reactivity to the 'novel tank' test.

In brief, 'reactive' zebrafish were observed to have higher whole-body cortisol than 'proactive' fish 6 min after exposure to the 'novel tank' test. Since 'reactive' and 'proactive' fish no longer differed in whole-body cortisol 30 min after stressor exposure, these differences at 6 min suggest that the groups differ in HPI axis reactivity. The authors also observed a sex difference in whole-body cortisol response, with males zebrafish having higher whole-body cortisol concentrations at the 6 min sampling time following stressor exposure.

While the overall aim of this study - to test whether differences in the behavioral response to stressors are underlain by differences in reactivity of the HPI endocrine axis - the manuscript in its present form should be revised to clarify several issues. Those issues are outlined below.

Major Issues (listed in order of importance for revision)

1) Baseline group. Individual zebrafish sampled at the 6 min and 30 min time points to generate the stress reactivity data were subsets of the High Stationary Behavior (HSB) and Low Stationary Behavior (LSB) artificially-selected lines. Those subsets were fish identified as showing consistent differences in depth preference in the 'novel tank' test (described in lines 118-126). Fish used for the baseline time point, however, were haphazardly selected from home tanks holding fish of the HSB and LSB lines (lines 131-133), and did not appear to be the same subset of fish classified as 'proactive' or 'reactive' based on the depth preferences. As such, the baseline fish are not a valid indicator of what baseline whole-body cortisol levels would be in the 'proactive' and 'reactive' groups, as they were likely to include a large scope of behavioral and (based on the author's hypothesis) neuroendocrine phenotypic variation.

Ideally, the authors would have used previously identified 'proactive' and 'reactive' fish for the baseline sample. Since this may not be possible at this time, the authors should revise Figures 1 and 2a to clearly indicate that the baseline fish are not equivalent to the 6 min and 30 min fish (perhaps by adding a dotted line between the baseline and 6 min samples, or gray shading the 0 min area of the plot). Similarly, the statistical analyses should be conducted for the 6 min and 30 min alone, and a separate analysis conducted for the two groups of baseline fish.

As commented by this and the other two reviewers, it is obvious that we were not clear in describing our lines and the use of the terminology of proactive and reactive groups. In our previous study (doi:10.1163/1568539X-00003018) we showed that differences in behavioral performance in the open field test (OFT) between the HSB and LSB lines are consistent across 5 other behavioral stress assays (including the novel tank diving test (NTDT) used here). That is, on average individuals of the LSB lines showed significantly lower amounts of stationary time in the OFT and significantly higher amounts of time in the top half of the NTDT compared to individuals of the HSB lines. From several other studies examining the behavioral consistency between and within individuals, molecular profiles, morphology of these lines (DOIs: 10.1163/1568539X-00003018, 10.1186/1471-2164-14-348, 10.1186/s12864-015-1626-x, 10.1038/s41598-018-30630-3, 10.1016/j.anbehav.2016.04.007), and endocrine responses from the current study, we believe that on average each line exhibit characteristics of either the proactive or reactive stress coping styles as described in the literature. Hence in the manuscript we call any individual from the HSB line as having the reactive stress coping style and any individual from the LSB line as having the proactive stress coping style; the coping style is a general characteristic of the line.

Despite the commonplace binary designation descriptor, it has been shown in the broader field of animal personality types that there is inter-individual variation in behaviors within a personality type (i.e. stress coping style). However, on average the two differ in their responses. We have shown that our lines are consistent with the observation of showing between-individual variation of behavior within a line but on average the two lines are significantly different from each other. Thus, we believe the proactive and reactive stress coping style descriptor is appropriate for any individual belonging to the LSB and HSB lines, respectively. We have included further description of our lines and justification in the use of the terms proactive and reactive stress coping style in the methodology (see lines 102-124).

The reviewer makes an astute observation regarding the baseline fish. The baseline fish may certainly represent individuals that do not have the same pronounced depth preferences in their respective lines. However, there is no feasible way to obtain such a comparable baseline group. As the reviewer pointed out, our methodology is such that we captured an individual from their home tank and immediately tested the fish in the NTDT. Only if the fish met the criteria stated in the methods did we sacrifice afterwards to examine the endocrine response. In other words, we did not pre-screen individuals beforehand; we also may not be able to pre-screen them in the NTDT as this would reduce the novelty stressor part on a subsequent test in the same assay. We would like to note that we also assayed a random assortment of individuals from each line and measured cortisol levels at 6-minute time point (see lines 128-131 of original submission). With those randomly selected individuals, we similarly saw (1) significantly higher cortisol levels in both the reactive ($t(22) = -3.122$, $p = 0.051 * 10^{-3}$) and proactive ($t(22) = -3.122$, $p = 0.005$) line compared to baseline and (2) that the reactive line had significantly higher cortisol levels than the proactive line at the 6 minute time point ($t(22) = 2.487$, $p = 0.021$). There was also no significant difference in cortisol levels between the randomly selected individuals and those behaviorally screened within each line at the 6 minute time point. We therefore believe that inclusion of the baseline timepoint in the statistical model with the other time points is valid. In the revised manuscript we note the possibility of baseline fish being behaviorally different from those at the 6 and 30 minute time point in the discussion but we believe it does not alter our results and interpretations (lines 224-229).

2) Clarity of HSB/LSB versus proactive/reactive phenotypes. The authors need to revise the manuscript to make the distinction between these phenotypes more clear to the reader. If I understand correctly, the genetic lines of zebrafish were artificially selected for HSB/LSB behaviors. The authors then are using those lines to identify individuals from each line showing 'extremes' of depth preference phenotypes following stress, and are referring to those phenotypes as "proactive" (for select fish coming from the LSB line) and "reactive" (for select fish coming from the HSB line). That needs to be made more clear to the reader in a revised version of the manuscript. I think that this could be done by revising the sentences describing the experimental design (Lines 82-93). Part of the confusion arises because the authors state that they are testing the HSB line versus the LSB line (see Lines 90-91, for instance), but are actually testing a particular phenotypic subset of the HSB line ("reactive" fish) versus a subset of LSB line ("proactive" fish).

We apologize for the confusion and have responded to this in our response to the first comment of this reviewer. In brief any fish from the HSB line is referred to as reactive and any fish from the LSB line is referred to as the proactive. We are not calling the extremes of each line as proactive or reactive. Rather they represent individuals of the proactive (LSB) and reactive (HSB) lines with the most pronounced depth preference. We have added substantial text describing the lines in the revised manuscript (lines 102-124).

3) Use of the term 'neuroendocrine.' At several places in the manuscript, the authors refer to their study examining "neuroendocrine responses" (see lines 24, 82, 199, 206, etc.). However, the current study looked at whole-body cortisol levels...which is not a neuroendocrine response, despite the HPI axis containing neuroendocrine signaling components such as CRH/CRF. This needs to be revised throughout the manuscript. The response variable measured in the present study should just be referred to as 'endocrine', given the data presented.

We have changed the wording through to the more accurate terminology of "endocrine" as appropriate.

Minor comments:

- 1) Lines 20-21. For clarity, perhaps revise to the following: "Animals experience stress in a variety of contexts, and the behavioral and neuroendocrine responses to stress can vary among conspecifics."
- 2) Line 28. Which behaviors? Since the specific behavior of depth preference isn't described in the Abstract, the reader has a difficult time evaluating the relevance of this manuscript to their own interests/research.
- 3) Line 30. The term 'cortisol' release isn't really accurate here, as these are whole body cortisol measures and not circulating. Hence, cortisol released from the interrenal is indistinguishable from cortisol produced but not released. Please reword.
- 4) Line 42. Please add citations here.
- 5) Lines 68-69. The point made in this sentence should be made more explicitly: that circulating cortisol concentrations may be influenced by several factors. Perhaps revise to: "Circulating cortisol concentrations also have been shown to vary with other factors including sex and recent behavioral experience, and such factors can contribute to individual variation in the temporal responses of glucocorticoids following the perception of a stressor."
- 6) Lines 73-74. Revise to: "Other closely linked traits may also be selected indirectly."
- 7) Lines 80-81. This sentence is worded awkwardly. Please revise.
- 8) Line 83. Revise "selected to display divergent" to "selected for divergent"
- 9) Line 105. The stressor here in the Novel Tank Diving Test (NTDT) is just moving the fish to a novel tank, is that correct? Perhaps make this explicit in this sentence for readers unfamiliar with this test.

We have made the suggested changes.

10) Lines 126-136. Please provide some number ($n = ?$) on how many fish were tested overall in order to identify the fish used for cortisol sampling (aka, how many fish from each the HSB and LSB lines were 'rejected' for not showing behaviors adhering to the "proactive" and "reactive" types as outlined in Lines 118-128?)

We tested a total of 44 reactive individuals and 219 proactive individuals. We have included the number of individuals that were "rejected" in the main text (line 151-152)

11) Lines 170-173. Were Pearson correlations run on all fish combined, or "reactive" and "proactive" groups separately? Or both?

We are grateful for the reviewer for this question. We originally just ran correlations with fish combined but not separately. When analyzing separately, we observed that there was a significant positive correlation between percent time spent in the upper zone and cortisol levels only in the

reactive line. All other correlations run on the lines separately were not significant. We have revised the text to include these results (lines 31-32, 200, 359-381).

12) Line 197. It's also possible that the reactive line shows a difference in glucocorticoid clearance rate. That should be mentioned as an alternative hypothesis here.

13) Line 206. The use of "neuroendocrine stress response" here should be revised to "HPA/HPI axis stress response", given that the present cortisol data and other studies being referenced refer to circulating glucocorticoids, and not hypothalamic-pituitary pathways.

We have made the suggested changes.

14) Lines 206-208. This claim that reactive individuals have a fasted cortisol release rate doesn't seem to be well supported given the present data. Can the authors back this up further? Otherwise, the statement is too speculative based on the data presented to this point in the manuscript and should be revised. However, the date from reference [42], appears applicable here. Please revise.

We appreciate the reviewer's comment to limit description and interpretation to only what we tested. As mentioned in response to another reviewer (see below), we did not originally intend our study to be a comprehensive time-series analysis of the endocrine response but had *a priori* motivations to select time points serving as a proxy for the rising and falling phases of the glucocorticoid stress response. We have now rephrased here and in other spots in the manuscript to be clearer that we just see significantly different cortisol levels during the rising phase of the glucocorticoid stress response.

15) Line 235. Delete the extra "be" between "further" and "modulated"

16) Lines 236-237. Revise to "...differences in the duration.."

17) Lines 238-239. This phrase about coping styles and different fitness optima is speculative, so should be worded as such. Perhaps revise to the following: "It has been proposed that discrete coping styles may be maintained in a population..."

18) Line 251. By "strain" do you mean "line"? I don't think this term has been used previously in the manuscript. For clarity, please use "line" here in Line 251 and in Line 301.

19) Lines 252-254. For clarity, revise to: "Our observation of higher stress-induced cortisol concentrations in males than females was not expected. Instead, female-based elevation of post-stressor cortisol levels has been seen across several taxa."

We have made the suggested changes.

20) Lines 292-293. Do males of the HSB and LSB lines generally show greater movement? Is this phenotypic difference seen even in the absence of partitioning HSB fish into "reactive" and "non-reactive" HSBs? Similar for LSBs?

We have previously documented that in an earlier generation of the proactive and reactive lines, males in both lines show greater movement than females.

21) Figure 1. The authors need to indicate whether these are mean +/- SEM values, or +/-SD for this and the other figures.

22) Table 1. It needs to be stated somewhere that these are p values being presented.

The figures are +/- SEM values and we have included that in the figure legend. We also indicated on Table 1 that the numbers are p-values.

Reviewer: 2

Comments to the Author(s)

I have completed my review of the manuscript titled “Differences in stress reactivity between zebrafish with alternative stress coping styles “ by Wong et al. The authors have executed a brilliant, but simple, experiment that demonstrates differences in cortisol reactivity between two coping styles. If I had one criticism, I would have like to see physiological data related to more parts of the HPA axis (e.g., gene responses of cortisol receptors). This would help to identify what parts of the pathway have been changed (perhaps the authors are currently working on this). I typically do not have any “negative” major comments, but I am at a lost to find anything serious enough to warrant any. I do, however, have some minor comments that the authors should be able to address in a timely manner. My recommendation is to accept with minor revisions.

Minor comments

- 1) Review manuscript to ensure the past tense is being used. For example, line 82, “we investigate ...” where further in the paragraph the past tense is used.
- 2) Line 83 – it isn’t clear to me the purpose. I think the confusing part of the sentence is the inclusion of the words after the colon.

We have made changes and modifications at these spots to improve clarity.

- 3) More details needed on the HSB and LSB lines. For example, how long have they been divergent (i.e., how many generations). What were the original parents (wild type)? What was the original selection experiment that assigned fish to the two lines?

We described the generation and maintenance of these lines in a previous study (doi:10.1163/1568539X-00003018). However we also feel that a more thorough description here is warranted given the confusion regarding the lines noted by all reviewers. We now include a more substantive explanation of how we generated and maintained the lines for the animals used in this study (lines 102-124).

- 4) 80 references is a bit much for a data focused paper.

We acknowledge the extensive reference section. Our intention is to give credit where credit is due and to put our results in the broader context of field.

- 5) Line 184 – should a reference be included?

We believe the placement of the references of that line is accurate.

- 6) The authors mention exposing zebrafish to a novel stressor. Unless I’m missing it, I didn’t see in the methods why was used as the stressor.

We apologize for the confusion. The NTD assay is itself the stressor (i.e. exposure to a novel environment). We have included a clearer description of this in the text (line 135).

Reviewer: 3

Comments to the Author(s)

In this manuscript, the authors quantified the whole-body cortisol levels in zebrafish that show different behaviours in novel tank diving test. The authors found differences in cortisol levels at 6 min between the animals that show „reactive“ vs. „proactive“ stress coping style as measured by the amount of time

they spent in upper vs. lower zones in a novel tank. The authors also measured the differences in male vs. female zebrafish in their cortisol response.

The study addresses interesting question of the relationship between neuroendocrine stress response and coping style. While the question itself is interesting, I do have several major issues with this manuscript.

1. The authors conclude „differences in cortisol levels between the alternative stress coping styles are not just restricted to magnitude but also extends to the temporal dynamics.“ However the authors only analysed 3 time points, 0, 6, and 30 min and showed that there is a difference only at 6. In so far as the temporal dynamics is the only thing that is measured in this manuscript, it would be important for authors to show more data points to demonstrate the actual temporal dynamics. They need to complete the time points at 12, 18, and 24 in addition to 6 and 30 min to show the peak values and the rate of decrease.

We appreciate the reviewer pushing us to think more critically about the terminology used to describe the conclusions we draw from our study. We agree that having additional timepoints will result in a more detailed view of the actual temporal dynamics and that the reviewer is inferring the use of the “temporal dynamics” descriptor is not appropriate for our manuscript. At the outset, our study was not designed to be a comprehensive examination of cortisol levels over time. Instead of examining the commonly used peak cortisol level time, we wanted to examine the cortisol levels during periods representing the rising (6 min) and recovery phases (30 min) of the glucocorticoid stress response in zebrafish. Hence, the inclusion of the additional time points would alter the original goal of our study. In our substantially revised manuscript, we have modified text throughout to make the goals and conclusions of the study more accurately aligned with each other. Additionally, we use more precise phrasing such as “changes in cortisol levels at time points occurring in the rising and falling phases of the glucocorticoid stress response” instead of “temporal dynamics” as appropriate.

2. One important question is whether different cortisol temporal dynamics shown in this manuscript is a general feature of selectively bred lines with different coping styles regardless of stressor type. Authors should test the temporal dynamic of the cortisol response of the lines to at least another stressor.

This is an insightful question by the reviewer. As with many studies examining cortisol stress responses within a population, typically only one stressor is used in each study. Our results certainly may only be limited to the novelty stressor and not extend to other stressor types. It is outside the scope of this study to examine whether the cortisol response observed here is restricted to the stressor used or if it is characteristic of all/other stressors (e.g. social stressor, food deprivation, chronic isolation, etc.). We hypothesize that the endocrine responses we observed are a general feature due to the observation of consistent differences in behavior between the lines across multiple assays. However, cortisol levels will need to be measured in response to other stressors. We have noted this limitation in our revised manuscript (lines 272-275).

3. How reactive and proactive lines defined by the authors other behavioural criteria (such as open field test) fare in the NTDT need to be shown. Do reactive lines always show significantly more bottom-dwelling behaviour compared to the proactive lines? For this study, the authors chose a reactive line fish if it spent greater than 151 seconds of the first 6 minutes. They also need to show the cortisol temporal profile of reactive line fish that may not show such a pronounced behaviour. Such a fish may classify as

“reactive” according to other behavioural criteria. In short a more comprehensive analysis is required to compare how reactive and proactive fish line behaves overall in NTDT test. Also the authors need to compare the cortisol response of different reactive and proactive fish and their behaviour in the NTDT not only the ones that show most pronounced behavioural differences in novel tank.

The reviewer asks important questions that we think also overlaps with concerns brought up by the other reviewers. We believe the confusion is due to our lack of thoroughness in summarizing what we and others have published regarding our lines and our use of the terms proactive and reactive. As noted in response to another reviewer we have generated two selectively bred lines of zebrafish starting from wild caught individuals. These lines were generated by selectively breeding groups of individuals displaying high stationary behavior in an open field test with each other and those with low stationary behavior together. We identify individuals that meet this criteria in each generation to produce the next generation (see doi:10.1163/1568539X-00003018 for more details). Hence, the fish used in our study have been selectively bred in this manner for 9 consecutive generations.

In our previous study (doi:10.1163/1568539X-00003018) we showed that differences in behavioral performance in the OFT between the HSB and LSB lines are consistent across 5 other behavioral stress assays (including the NTDT used here). That is, individuals of the HSB line displayed more risk-averse behaviors relative to individuals from the LSB line across 5 other assays. More pertinent to the reviewer’s point, individuals of the LSB lines showed significantly lower amounts of stationary time in the OFT and significantly higher amounts of time in the top half of the NTDT compared to individuals of the HSB lines on average. We refer the reviewer to this previous study for a more thorough characterization of the behaviors of both lines and sexes in the NTDT (and 5 other behavioral assays). From several other studies examining the behavioral consistency, molecular profiles, and morphology of these lines (DOIs: 10.1163/1568539X-00003018, 10.1186/1471-2164-14-348, 10.1186/s12864-015-1626-x, 10.1038/s41598-018-30630-3, 10.1016/j.anbehav.2016.04.007), we have multiple lines of evidence to suggest that the LSB and HSB lines exhibit characteristics of the proactive or reactive stress coping styles, respectively, as described in the literature (*sensu* DOIs: 10.1016/j.yfrne.2010.04.001, 10.1016/j.tree.2004.04.009, 10.1016/j.neubiorev.2006.10.006). Hence, in the manuscript we call any individual from the HSB line as having the reactive stress coping style and any individual from the LSB line as having the proactive stress coping style. As we have shown in other studies and in the data seen in the current manuscript, our lines certainly have variation within a line. However as mentioned above, on average the two lines differ in numerous behavioral, molecular, and morphological characteristics consistent with the either the proactive or reactive stress coping style. We have added substantial text that further describes our previous studies using these lines and how these lines are generated and maintained (102-124).

The fish chosen for our data represent the “extreme” ends of the spectrum within each of our HSB (reactive) and LSB (proactive) lines. As mentioned in the methods, this was done under the assumption that these individuals would have the most robust glucocorticoid responses. We appreciate the reviewer’s suggestion that the results observed here may only be limited to those on the extreme end of each line or perhaps represents a state response (rather than a trait). We too were concerned of this possibility. As such we tested individuals of each line that did not undergo any time criteria (they were randomly selected) for 6 minutes in the NTDT (see lines 224-229). We still observed a significant difference in cortisol with the reactive (HSB) individuals having significantly higher cortisol levels than the proactive (LSB) individuals. Further these individuals did not show any significant difference in cortisol levels with those who were behaviorally screened at the 6 minute

time point. Hence, we have evidence to suggest that the glucocorticoid responses are a general trait of the line and not representative of a subpopulation.

4. Somewhat related to the point 3, both in abstract and in the text the authors mention that they couldn't find correlations between individual variation of cortisol levels and behaviour, but this important point is not shown. The authors need to show this data and address the point of individual variation.

As mentioned in reviewer #1 response, we originally ran the correlations on both lines combined. At the suggestion of reviewer #1 we also ran correlations on the lines separately. When analyzing separately, we observed that there was a significant positive correlation between percent time spent in the upper zone and cortisol levels only in the reactive line. All other correlations run on the lines separately were not significant. We have revised the text to include these results (lines 31-32, 200, 359-381). We also now include a figure of these results (Figure 4).

Appendix B

Response to Reviewer Comments

ID: RSOS-181797.R1

Dear Editor

We are grateful for all of the constructive feedback from yourself and the reviewers throughout the process. We have made the minor corrections as suggested.

Sincerely,

Ryan Wong, Jeffrey French, and Jacalyn Russ

Reviewer comments to Author:

Reviewer: 2

Comments to the Author(s)

None

Reviewer: 1

Comments to the Author(s)

Overall, the authors made appropriate revisions to the manuscript, and the major issues that arose with the prior version of the manuscript have been addressed. I have only a few small corrections for the authors with the revised version of their manuscript:

Lines 82-83. Please list for common names for each taxon here (Japanese quail, rainbow trout)

Line 236. 'great tits' shouldn't be hyphenated.

Line 371. Revise to "suggests"

Line 386. Revise to "reactive line"

We have made all of the suggested changes.